# Graphene-Enhanced Polydimethylsiloxane Patch for Wearable Body Temperature Remote Monitoring Application

**DOI:** 10.3390/s22239426

**Published:** 2022-12-02

**Authors:** Jie Huang, Daqing Huang

**Affiliations:** 1College of Electronic and Information Engineering, Nanjing University of Aeronautics and Astronautics, Nanjing 210024, China; 2School of Aeronautic Engineering, Nanjing Vocational University of Industry Technology, Nanjing 210023, China

**Keywords:** graphene, temperature monitoring, thermal conductivity, wearable devices, polydimethylsiloxane

## Abstract

In this work, we designed and implemented a wearable body temperature monitoring device, which was constructed by a graphene-enhanced polydimethylsiloxane patch and a temperature measurement chip. The body temperature patch adopts a completely flexible solution in combination with near field communication component, which provides the advantages of passive wireless, overall flexibility, and being comfortable to wear. The whole device can be bent and stretched in conformal contact with skin. In order to improve the temperature conduction ability of the patch and make the patch data more accurate, we adopted graphene nanoplates to improve the thermal conductivity of polydimethylsiloxane patch with a significant thermal conductivity increase of 23.8%. With the combination of hollow sandwich structure and small dimension. it will reduce the uncomfortable situation of wearing the device for extended periods and can be served to monitor the human body temperature for a long time. Ultimately, this device is combined with a reading software for analyzing and processing on a smart mobile terminal. The real-time and past temperature range can be a pre-warning; meanwhile, the historical data can be traced and analyzed. Therefore, this device can be utilized in multiple human body temperature measurement scenarios and complex public health situations.

## 1. Introduction

After a long evolution, the creatures on the earth have undergone the evolution from cold-blood to warm-blood. This evolution helps warm blood creatures to widely expand the space and time for activities, for the body temperature regulation function of warm-blood animals can ensure a relatively constant temperature. Hence, the constant body temperature can help warm-blooded animals better adapt to temperature changes [1]. There is no doubt that this advantage has a significant meaning to human beings during their evolution. On the other hand, body temperature can also reflect the status of creatures, including activity and health status. When the body temperature fluctuates abnormally, people need to find out the reason and take measurements of the temperature in time. Otherwise, it may cause serious damage to the body, for example, the human body with prolonged exposure to high temperatures may cause irreversible damage to the nervous system [2]. At present, most body temperature measurements are based on intermittently sampling and analysis of the unbalanced body temperature. However, the continuous and real-time measurement of body temperature is still required. For example, patients in hospitals require medical staff to measure their body temperature every three to four hours after operation [3]. The patients required continuous monitoring of body temperature, as an important index to evaluate the patient’s health condition. Newborns also need continuous temperature monitoring, for their poor immunity and insufficient self-temperature regulation ability [4]. It is necessary to continuously monitor the body temperature and take immediate measures to get involved with body temperature management to ensure the human being’s health [5]. The body temperature can also be introduced to evaluate our biological clock and basis of metabolic rate, which can be used to guide daily work and rest, and analyzing athletes’ sports performance [6]. In the field of body temperature measurement, traditional thermometers (e.g., mercury/alcohol thermometers) utilize the principle of thermal expansion and contraction of gas or liquid [7]. Other thermometers are mainly based on thermocouples and thermoresistance. With the development of modern technology, thermometers also evolved in various forms, including hand-held heat radiation thermometers and body temperature patches [8]. The common temperature measurement sites in medicine are fundamentally measured at underarm, cochlea, oral, and rectum, among which the rectum-type is the most accurate. In addition, the underarm temperature is most commonly utilized for its convenience [9]. Therefore, to develop a type of remote and wireless online body temperature patch is extremely urgent.

In recent years, some outburst public health events have caused tremendous hazard to people’ health all over the world. According to previous studies, during the epidemic prevention process, more than 98% of infected people had clinical manifestations of hyperthermia. Thus, the body temperature measurement is one of the main approaches in epidemic prevention and widely applied in many controlled areas. At present, the mainstream of temperature measurement method still relies on an infrared thermometer to measure the temperature of bare skin. This method has several drawbacks in many aspects. When the environment temperature is cold, the temperature of bare skin will be lower than the body’s temperature. Moreover, during the temperature measurement, medical staff needs to be in close contact with the patient, which increases the risk of exposure infection. In response to this problem, this work proposes a wearable body temperature monitoring device, which is flexible, stretchable, and bendable in its texture. All these mechanical properties ensure it is comfortable to wear on the body for long duration. The passive wireless component in this device can be connected to smart mobile terminals for data reading and uploading with near-field communication (NFC) technology [10]. This wearable body temperature monitoring device can be potentially applied in many events, such as personal health care, epidemic prevention and control, body living status monitoring, long-term measurement of newborn body temperature, sports performance analysis, and many other fields.

## 2. Materials and Methods

### 2.1. Design Outlines

According to the above research and analysis results, it is not difficult to find that a flexible, passive wireless, stretchable, and bendable flexible body temperature patch that can be connected to intelligent terminals for data uploading has not yet been developed, and such a body temperature patch has a special value at present:For personal health care, body temperature is closely related to the biological clock. Long-term monitoring of body temperature helps users to improve the sleep quality, diet regularity [11], etc. On the other hand, female long-term monitoring of body temperature helps to determine whether the ovulation period is abnormal, in order to remind them go to the hospital for examination.For patients, continuous temperature monitoring during illness and hospitalization is helpful to judge physical recovery [12], and monitoring the temperature change curve of patients is also helpful for doctors to make more appropriate treatment plans [13].For infants and young children, body fever often appears during the night. If the fever has not been noticed for a long time, it may cause serious consequences. Thus, stable and reliable access to infants and young children’s temperature is necessary. Meanwhile, for baby’s tender skin, the comfort level of the thermometer or temperature patch is also worth paying attention to [4].For athletes and other professional fields, the ability to monitor body temperature without affecting athletic performance during exercise is critical to analyze the actual metabolism and real-time status of the body, adjust the off-site training plan, and reasonably distribute stamina consumption on the field [14].The large-scale sudden events of infectious diseases occur on the recent domestic and international regions. The main symptoms of the majority of patients are fever and hyperthermia (over 98%) [15]. Thus, for the epidemic control and prevention, the preliminary healthy intervention is one of the primary effective methods for measuring temperature. Thus, the history of the previous few days temperature data is also important. They need a convenient method of temperature measurement and historical data reading approach: non-contact, low cost, convenient measurement (within 1 s), not affecting daily life and travel, convenient large-scale production. It is also the main design target of this project.

### 2.2. Materials of Patch

Currently, the domestic Internet of things (IoT) industry is developing rapidly, and many large enterprises are entering in this field continuously. In the whole network, NFC, the near-field communication mode, has its unique advantages, such as its short communication distance, which can ensure certain security from the hardware, and the mature technology with low cost. Many smart phones and other terminal devices are equipped with NFC reading and writing function, which is the premise of large-scale application of NFC. Therefore, it is taken as the communication method of temperature patch and other functions to achieve data reading, writing and analysis. Such processing also ensures a certain openness, and it is possible and operable to access more health systems in the future [16]. On the other hand, to realize the large-scale sudden events of infectious diseases warning, it is also feasible to write more identifying information and ID tags, which can be further read out historical data into the cloud.

The basic components of polydimethylsiloxane (PDMS) are viscous. Therefore, it requires the addition of dopants to the curing agent before making PDMS solid. The as-received PDMS was purchased from DOWSIL^TM^ (DC-184) and the dopant was optimized to mix within 10 wt.%. After fully mixed with curing agent, the PDMS was sent to dope graphene plates to enhance its thermal conductivity, according to the principle of solid thermal conduction [17]. If it increases the content of electrons or phonons, it hopes to increase the thermal conductivity of a certain material [18]. On the other hand, the flexible materials in this work are used as packaging for hardware circuits. Thus, they require electrical insulation in order to avoid the local or global conductivity of the doped materials connected within the doped graphene nanosheets. However, the metal conductive materials will not be suitable for doping. From this perspective, insulating and high thermal conductivity materials, e.g., graphene nanosheets [19] and diamond nanoparticles (ND) are the first choice, for their unique properties of thermal conductivity, and biocompatibility [20]. Thus, different types of doping with additives (pure PDMS, 8.7 wt.%, 12 wt.%, 17 wt.% and ND addictive) in PDMS were attempted (shown in Figure 1a–e). The graphene flakes were first dispersed uniformly in a curing agent by a 20 min ultrasonication bath. The average lateral size of graphene flakes varies from 500 nm to 300 μm according to our previous research [21]. However, the size distribution of graphene flakes was difficult to define, for they always have significant aggregations. The layers of graphene flakes can be deduced from Raman spectra in the next section.

### 2.3. Optimization and Characterization

However, the dosage (per unit volume) of dopant that can match the curing agent is limited. In the experimental process, the following proportions are determined. In the first process, a dopant agent (17 wt.%) has been added to the basic components of the PDMS for curing. The dopant will form a uniform distribution in the PDMS along with the curing agent. As a dopant, it is worth noting that graphene nanosheets as a dopant wrapped in PDMS materials will not cause a short circuit in the electrical path. The thermal conductivity meter model Hot Disk TPS 2500S instrument was utilized to measure its parameter. According to the transient plane heat source method, the thermal conductivity of the samples can be directly measured. Finally, the fifth group doped with 17 mg was selected as the optimized one. The thermal conductivity of the sample increased the most, which was 27.1% higher than that of the first control group, from 160.3 mW/(m·K) to 203.7 mW/(m·K). Considering the uniformity of the distribution and its effect on the original properties of the final product will no longer increase with the doping ratio, the graphene nanosheets were selected as the dopant.

After the optimization and the improvement of the material’s thermal conductivity, the production of the body temperature patchs are fabricated as follows: First, dump doped PDMS (1:20 with curing agent) on a glass substrate. Then, take it out after heating it at 75 °C for 45 min. Next, the welded hardware is placed and adjusted the circuit to ensure it is in perfect contact with the first layer of PDMS. Subsequently, the substrate was heated at 75 °C for another 65 min for the second time consolidation, and then remove from the mold. After a 30 min cooling process, the surface of the substrate has a matte texture. Then, coat a layer of doped PDMS on the substrate. Finally, it is annealed at 75 °C for 2 h, and completely take it out of the other mold. In the above steps, it is worth noting that when burning the program into the chip applied, it requires the use a self-made burning board and it should be taken out after the annealing program. The circuit integrates a NHS3100 chip, which includes NFC function and temperature sensor function, which can make the detecting circuit more concisely.

Figure 2 presents the microscope images of the fabricated PDMS patch after graphene enhanced and the characterizations. Figure 2a shows the inverted pyramid structure in PDMS patch after peeling it off from the mold and is completely annealed. For the higher magnification, the detailed inverted pyramid structure was demonstrated in Figure 2b. The SEM images are snapped by Opton COXEM EM-30N. It is observed that there are scarcely any cracks and contaminations on the pyramid structures. For SEM observation, the graphene in PDMS does not have a significant difference in the images for it distributes homogeneously inside the PDMS bulk after fully annealing, and does not appear on the surface. The graphene nanosheets observed under high magnificent images were acquired from FEI Titan 80–300 keV high resolution transmission electron microscope (HRTEM), as shown on Figure 2c, while the fast Fourier transform (FFT) pattern is inserted in the inset figure. It was observed that the graphene was constructed in a planar honeycomb network. Meanwhile, the unit cell of single-layer graphene consists of two carbon atoms with a distance of 1.42Å and lattice constant of 2.46Å [22]. The Raman spectra of the pristine graphene nanosheets (Figure 2d) was acquired from HORIBA XploRA PLUS with a 514 nm wavelength. The *G*-band peak was spotted at 1582 cm^−1^ and 2*D*-band located at 2691 cm^−1^, which was assigned as the planar configuration of *sp*^2^ carbon of constituted graphene and 2*D*-band, respectively. According to the Raman theory, the layer of graphene flakes can be approximately determined as no more than 10 layers by calculating the ratio of *G* and *2D* band [23].

The patch designed in this work has the following advantages: It can be used on most parts of the human body, such as the chest, underarms, forehead, behind the ears, etc. Therefore, the size of the patch is required to be within 3 mm in thickness, limited to 4 cm in length and width, and is overall flexible and conformable to the skin [24]. The patch needs to be waterproof to prevent the internal circuit from being corroded by sweat, thus, the circuit needs to be encapsulated. Since the requirements for the upper and lower packaging materials are slightly different, and the middle circuit part needs to be waterproof and mechanically protected, the entire patch can be attached. The temperature patch was designed as a sandwich structure, divided into upper, middle and lower layers. The upper layer is a waterproof protective layer, the middle layer is a circuit part, and the lower layer is a skin contact layer, which also has a waterproof function. The upper and lower layers meet at the edge to seal the entire circuit in the packaging material. The use of body temperature patches will inevitably cover some of the pores, and patients with high temperature are more likely to sweat. To prevent the patch from slipping due to sweat, the patch can be hollowed out. Digging holes in the patch area will reduce the influence of sweat on the adhesion ability of the patch by reducing the coverage area. Until now, our group has fabricated and integrated over ten types of other sensors with the patch, and all of them shows excellent repeatability during the test.

### 2.4. Electronics Configurations

Figure 3 represents the schematic diagram of electronic configuration on graphene-enhanced PDMS patch for wearable body temperature monitoring. To prevent the patch from adhering to the clothing, the protective layer of the patch is required to be smooth and non-sticky, while the chip pins of the middle circuit layer are directly soldered with copper wire. It has higher requirements on the reliability of solder joints [25]. At the same time, the serpentine line strategy is appropriately utilized so the circuit part can share the tensile force under stress and enhance the stretchability of the patch. Considering the chip size, the difficulty of soldering and the thickness of the copper wire, this part of the circuit can utilize the QFN24 packaged chips. The lower layer material should have adhesiveness to allow the patch to stick onto the skin surface layer. At the same time, this layer should be connected to the upper layer to form an encapsulation layer in order to protect the middle layer. Thus, ensuring the waterproof performance and to make the circuit structure more stable. According to the above analysis, the thickness of the whole patch can be controlled within 3 mm.

The NHS3100 is an integrated chip optimized for temperature monitoring and recording with an embedded NFC interface, internal temperature sensor and direct battery connection. It is an effective system solution with minimal external components and single-layer foils for temperature monitoring. Its HFQFN24 package size is: 4 × 4 × 0.85 mm, the accuracy can reach 0.1 °C, the external NFC antenna can achieve realize the function. Figure 4a is the pin configuration diagram with the HVQFN24 package of NHS3100 chipset, while Figure 4b is the power supply of the temperature patch. Figure 4c shows the overall schematic diagram of NHS3100 chip. It consists of the power management unit, clock, timer, digital computing and a control cluster (ARM Cortex-M0+ and memory), Advanced High-performance Bus (AHB) and Advanced Peripheral Bus (APB) slave module. The NHS3100 supports various power control functions in active mode. When the chip is running, power consumption and the clock of the selected peripheral can be optimized to reduce power consumption. There are also three special processor low-power modes: Sleep mode, Deep Sleep mode, and deep-power drop-down mode. However, because the Radio Frequency Identity (RFID)/NFC communication module is in self-powered mode, the basic RFID/NFC tag functionality can be maintained in all sleep and power drop-down mode. This chip has a variety of communication interfaces, supports an NFC/RFID ISO 14443 standard type interface. It is compatible with NFC 2 standard interface, at the same time its I^2^C bus interface. This chip supports the complete I^2^C bus specification and fast mode, data rate is 400 kbit/s, with multi-address identification and monitoring mode. There is an 8 MHz RC oscillator inside the chip with an accuracy of 2%. The power supply mode of the chip is flexible, supporting external voltage from 1.72 V to 3.6 V. It can also be powered and activated through NFC field, and the internal integrated power management unit (PMU) is used for general power consumption control. The ARM Cortex-M0+ supports four low-power modes: sleep, deep sleep, power-off, and deep power-off. At 3.0 V battery off mode, the chip current consumption is less than 50 nA. In addition, the chip has a unique device serial number, which can be read and identified.

The NFC module in the chip is utilized to control the communication between the chip and the reading device in order to obtain energy from the RF field. The module in the chip is compatible with most NFC standards, and the internal capacitance is 50 pF, which is compatible with standard NFC antennas. The RFID/NFC interface allows communication using 13.56 MHz proximity signaling. The RFID interface operates internally at 6.78 MHz. This clock is recovered from the RF field and is independent of the NHS3100 system clock.

The temperature sensor is integrated in the chip and the high precision scaled analog-to-digital converter (ADC) is used to measure the temperature, which can be accurately measured in a wide range (−40 °C–+85 °C) and higher accuracy will be achieved in 0 °C–40 °C. Since the temperature sensor in this topic is mainly applied for measuring human body temperature, the accuracy requires to reach 0.1 °C. Through the preliminary calibration, the chip temperature sensor can achieve the same accuracy as the existing commercially available thermometer.

Due to the patch requirements of overall flexibility, the printed circuit board is no longer suitable. This research used copper wires to directly connect the chip pins with serpentine wires to increase the circuit stretchability [26]. The whole wearable patch was designed as shown in schematic drawing in Figure 5a–c. The detecting sensor, including chipset and its connecting NFC coil (see Figure 5d) was fully covered by a graphene-enhanced PDMS layer. The whole patch was hollowed in post-optimization procedure with inverted pyramid structure by stamping technology (Figure 2a,b), in order to increase the comfortability and performance onto the skin. The whole packaged sensor and patch can be attached on the body conformally (Figure 5e).

The experimental area is the back of hand after cleaning and disinfection. Each paste was tested to peel off many times, and each cycle repeated 100 times. After 100 cycles, the paste can still stick on the back and will not be peeled off by regular body movement. However, as a result of 100 times repeated process, it sticks some dusts on the surface patch. It can be cleaned up by using hand sanitizer and alcohol sanitation after repeating peeling-off and attaching 100 times. It was found that in the first 700 times experiment, the patch consistently performed successfully and used as in normal. After 755 times, the edge of the PDMS package began to crack and the data could not be read due to the fracture of the welding point, which was the result of defects in the production process and repeated tearing. This test verifies the excellent repeatability of the patch.

## 3. Modelling and Simulation

As a body temperature patch, the precision of temperature detecting is the priority. After improving the comfortability and thermal conductivity of the PDMS patch, its heat transformation behavior and flexibility were simulated through the COMSOL software, as shown in Figure 6a–f. We took measurements to explore the thermal equilibrium state of the patch in these two scenarios. According to this division method, it can essentially cover most medical and daily temperature measurement scenarios. The simulation is performed and the heat transfer module in the software has been utilized to simulate and reproduce the scenario.

### 3.1. Temperature and Stress Modeling

Figure 6a shows the temperature measurement simulation of the bare skin area. The room temperature is set to 22 °C, the skin temperature is set to 36 °C, the patch length is 35 mm, the width is 25 mm and the thickness is 3 mm. According to the temperature distribution in the figure, it is obvious to find that the patch has large temperature gradient in the *z*-axis direction, which is due to the low thermal conductivity of the flexible material. In view of this temperature gradient distribution, the temperature sensor can be placed close to the skin to obtain accurate body surface temperature data.

In order to more accurately simulate the state of the patch in the area of cloth covering on the skin, the simulation also added surface air flow to simulate the temperature state of the patch when the air flows through (Figure 6b). According to the simulation diagram, the temperature difference between the upper and lower layers is extremely tiny (upper layer is 35.9 °C, lower layer is 36.5 °C close to the skin, the patch length is 35 mm, the width is 25 mm and the thickness is 3 mm), so the whole patch is in a state of temperature equilibrium and this state is highly ideal. Due to this, the position of the temperature sensor has many choices for placement, rather than being placed in a specific location.

In addition to thermal analysis, the patch requires to face a variety of deformation, stretching and bending in the process of production in daily use. The requirement of demolish in the process of applications will produce deformation and stretching in together with the human skin, which has further requirements on the overall mechanical structure of the patch. The simulation results are demonstrated as Figure 6c. The risk of whole patching process mainly derived from internal and external circuit encapsulation materials elastic modulus differences. The same stress force could lead to a three-layer structure have different deformation, further lead to the split between layer and layer separately. It requires to determine the scope of the deformation to improve the patch if necessary. Through the simulation results show that internal force of 5.13 × 10^−3^ Pa longitudinal stress will not have huge effect on the patch. Based on the characteristics of flexible material utilized in the packaging, it is a feasible way to share the stress and will not produce huge local force. The significance of this simulation result is confirmed that patching the stability of the mechanical structure. In addition, considering the three-layer structure of the patch, it tends to cause deformation and eventually damage the patch due to the mismatch of thermal expansion coefficient. Therefore, corresponding simulation should be carried out to investigate the stress distribution caused by internal deformation after heating.

When the patch is subjected to frontal pressure, if the pressure distribution is uneven, the internal circuit will directly connect to the chip pins with copper wires. The solder joints are suspended and excessive force on the internal circuit solder joints may cause a circuit short, affecting the normal function of the circuit part. Due to the package is difficult to repair, it is necessary to perform a stress simulation analysis in advance to eliminate potential risks. Figure 6d shows the simulation diagram of temperature measurement in bare skin area. It can be found that because of the flexible material package, the entire patch is uniformly stressed and the effect is ideal. Further experiments can be conducted to verify the pressure-bearing capacity of the patch structure. In addition, because the patch contains both flexible packaging materials and hardware circuits, when the patch has been heated, the deformation of the patch will be different due to the difference in deformation of different parts. 

According to the simulation diagram, due to the existence of a temperature difference, the temperature gradient of the patch is extremely low and the temperature in the *z*-axis direction shows a linear distribution (Figure 6e). The main reason for this phenomenon is that the packaging material utilized in this work is flexible and its thermal conductivity is severely low, normally ~100 mW/mK (150 mW/mK was applied in this simulation). The guiding significance of this simulation result lies in the location of the sensor. Due to the existence of temperature gradient in the *z*-axis, the temperature sensor should be as close as possible to the skin, so that the temperature measured by the sensor will be closer to the skin temperature in the actual use process. On the other hand, the existence of such a temperature gradient is not the worst, because it can play a certain insulation role. In the same case, the surface temperature of the patch covered area is closer to the core temperature of the body, which is relatively seldom affected by the temperature difference of the outside environment.

Figure 6f simulates the deformation and internal stress distribution of the patch after heating. The internal maximum stress is 2.9 × 10^5^ N/m^2^, which appears in the circuit layer. Another lower layer has a minimum of 2.0 × 10^5^ N/m^2^, which exists on the packaging material layer. The thermal expansion coefficients of the materials are different. The thermal expansion coefficient of copper in the circuit part is 1.67 × 10^−5^ m/K, and the thermal expansion coefficient of the flexible material for packaging is 1.23 × 10^−1^ m/K, which causes stress existing on the circuit part. The influence of stress intensity on the patch is limited and the probability of the patch will not collapse and further verification will be carried out in subsequent practical utilization.

In the previous thermal conductivity simulation, it was found that because the thermal conductivity of flexible materials is generally low, the thermal conductivity required to be improved when applied to the field of temperature measurement. Some researchers have made significant effort before [27], such as doping, recrystallization, etc. Comprehensively considering the materials and operability used in this work, we decided to apply the mixing method to improve the thermal conductivity. The thermal conduction mechanism is different for different materials [28]. 

From the perspective of the material itself, it can be divided into two categories: crystalline and amorphous. The thermal conductivity of a crystal comes from the vibration of phonons and free electrons generated by the lattice vibration, while the thermal conductivity of a non-crystal is provided by the thermal vibration of atoms or molecules. The disorderly arranged atoms or molecules in the non-crystal vibrate around a certain point, transferring energy to adjacent atoms or molecules. On the other hand, amorphous can also be regarded as especially fine-grained crystals, so the concept of phonons can also be introduced when analyzing its thermal conductivity. However, for this part of the material, the free path of phonons is extremely short, therefore its thermal conductivity is terrible. Both graphene and diamond nanoparticles have ultra-high thermal conductivity. The thermal conductivity of graphene comes from the contribution of different phonons and their unique structure. For diamond, it is due to the smaller mass of carbon and stronger carbon–carbon bond. These properties make the propagation of vibration in diamond extremely smooth. Through the above analysis, we tried to add materials with high thermal conductivity to materials with low thermal conductivity to increase the phonon or electronic thermal conductivity of the material, thereby improving the overall thermal conductivity. However, mixing graphene or nanodiamond also has problems, such as how to do uniform distribution of mixed materials, mixing or doping ratio, and mixing approaches will affect the original mechanical properties of the material.

### 3.2. Simulation of Doping in PDMS

In this work, after comparing many existing flexible materials, polydimethylsiloxane (PDMS) is selected as the encapsulating material, the thermal conductivity is 150 mW/(m·K) and the mixing materials are graphene nanosheets and diamond nanoparticles in order to analyze the thermal conductivity and mechanical properties before and after mixing [29]. To verify whether mixing can improve thermal conductivity, we introduced the heat transfer module of COMSOL software for simulation. Figure 7a,b are the thermal conductivity diagram and thermal isotherm before doping. Figure 7d,e are the thermal conductivity and heat conduction isotherms diagram after doping, while Figure 7c,f represents the stress analysis distribution before and after doping. Due to the amount of doped powder particles is extremely large, only qualitative simulation studies can be carried out here, and the optimized doping dose and effect are required to be further verified by experiments.

Due to the restriction of the simulation software itself, the main purpose of this simulation is qualitative research, mainly exploring the feasibility of improving thermal conductivity through doping. According to the simulation results, when the material size is the same, the temperature gradient is the same. The thermal conductivity of the doped PDMS is improved to a certain extent and the temperature distribution is uniform. Therefore, the improvement effect of doping on the thermal conductivity can be further determined through experiments. In addition to the simulation of the shift of thermal conductivity, the dopant is not a flexible substance. Considering the overall stress process of the material in the twisted state, it is necessary to additionally verify its mechanical properties after doping. After applying the surface stress, stress non-uniform material within the overall, doping does not have significant influence on mechanical properties of PDMS. However, considering copper doping dose at the same time, every time the number of nanoparticles doped is a huge, simulation software that is difficult to achieve.

According to the principle of solid conduction, as long as the content of electrons or phonons increases, it is hopeful to enhance its thermal conductivity [30]. On the other hand, the flexible materials in this article are mainly used for the packaging of hardware circuits, so they need to have insulation. In order to avoid the local or global conductivity of the doped materials connected within the doped object, the metal conductive materials will not be used for doping. From this perspective, insulating and high thermal conductivity materials such as diamond nanoparticles have become the first choice. By comparison, taking into account the availability of materials, doping effects and other factors, this paper selected graphene nanosheets and diamond nanoparticles [31]. As a dopant, it is worth noting that graphene nanosheets as a dopant wrapped in PDMS materials will not cause a short circuit in the circuit. After comparing the changes in thermal conductivity between the two, it is preferable to use them. On the other hand, because the patch is the means of measuring body temperature, thus the heat conduction performance is also a topic of focus, can be found by the preceding analysis as the thermal conductivity of PDMS is extremely low. This is because the PDMS’ internal neither free electron thermal conductivity (electronic freedom movement), there is no orderly arrangement of the lattice (phonon heat conduction vibration). In the previous section, we have analyzed how to improve the thermal conductivity of amorphous materials. Increase the content of metal or crystal parts inside the material to provide additional electrons/phonons for conducting electricity [32], so this problem can be solved by doping, for example, graphene nanosheets [33], diamond nanoparticles and other high thermal conductivity materials are incorporated in the process of preparing flexible polymer materials. However, there are also some problems in the doping process, such as how to make the doping materials uniformly distributed, which will be tried and solved in the following experimental process [34]. The first step is to conduct qualitative verification of this idea, to verify whether doping of PDMS can improve the thermal conductivity of materials and whether doping will affect the mechanical properties of materials (mainly on its tensile and bending properties) [35].

## 4. Results

Figure 8a shows the screenshot of the software logo and the main interface of the software. The screenshot of the software starting page and the screenshot of the temperature reading page from left to right, respectively. The functions of the software are temperature interval alarm, automatic reading time interval setting, etc., which will show in most of the body temperature measurement scenarios. After opening the APP, the NFC reading area of the mobile phone should be placed close to the temperature patch, while the ID and current temperature data of the temperature patch can be read out. If power supply or battery needed, it only needs to connect to the button battery and power protection circuit. It can write data to the chip at the same time, set the temperature reading intervals. Figure 8b shows the scenario of the main entrance of university and Figure 8c is in the classroom with convenient Android smartphone temperature reading. The temperature patch can be reused several times after peeling-off from bare skin (Figure 8d,e).

The domestic schools, community, entrances of buildings often require temperature measurement and identifications [36]. The entrance guards always check the temperature by infrared(IR) thermal images or IR thermometer guns. People can combine the two approaches to do authentication and temperature reading all at once. At the same time, the historical temperature data in the patch can be read to further judge the health status of the patch user and prevent the case of missing checks in patients with intermittent fever.

Normally, the body temperature before and after sleep will obviously fluctuate up and down. By monitoring temperature increase or decrease and provide a relationship between the two will help people to fall asleep quickly, and properly wake up later when body temperature rises, which will help people stay awake after sleep and reduce discomfort. Meanwhile, the whole sleep cycle is monitored to determine the quality of sleep and provide reasonable sleep recommendations.

For athletes and ordinary people, monitoring the changes of body temperature are necessary [36]. Due to body temperature and metabolism efficiency having a positive correlation, such as the human body temperature in swimming pool water while continuously producing heat. Symptoms may result in a loss of temperature, which can make a person lose activity, or even experience heart failure, which is extremely dangerous in the water. In addition to the temperature change during the process of movement, if the exercise intensity is excessive, the movement after excise continues to burn calories, which is called after-burning. It will continue to burn calories after the body would have been in a state of burning fat. Meanwhile, the metabolic rate is much higher than the usual average temperature rise. The subjects in the test can feel a slight warmth in the muscles, often accompanying with a slight state of sweating.

Figure 9a–h are the temperature recording curves of the author of this study for 48 h. The circular direction of the figure is the corresponding time, and the concentric circles are the temperature read-out scale. Small icons correspond to sleep onset and mealtime. The subject under test followed basically the same routine in those days: go to bed at 02:00 and waking up at 9:30 with no rest in between. It is obvious that within two days, the body temperature of the first 24 h and the last 24 h followed basically the same change trend. The body temperature went into a low level at 02:00 every day after falling asleep, and this state lasted until 18:00 of the same day. This is because the subject under test did not eat normal food at noon, and only consumed 180 mL milk and 150 g whole wheat bread, which leads to the body metabolism staying at a low level. After recover eating at 18:00 p.m., body temperature begins to rise and the metabolism enters a higher level until sleep returns. It can be preliminary inferred that if individuals follow the same sleep and eating habits for a long time, their body temperature might also have similar changes and interact with the circadian clock.

The subjects have conducted experiments to verify it and the temperature record is shown as Figure 9a–e. This figure represented body temperature data collected from the test subjects under the situation of staying up late within 12 h. The gray inner circle represents the sleep state, from 03:20 a.m. to 10:40 a.m., while the inner circle of the circular line is 36 °C and the outer circle is 37 °C. This temperature range was divided into ten equal parts in the middle. Each cell represents 0.1 °C and the blue broken line is the temperature data. It is not difficult to find that the body temperature before bedtime fluctuates around 36.7 °C and occasionally fluctuates to 36.8 °C. When the people begin to sleep, the body temperature usually has a corresponding decrease. After falling asleep, the body temperature drops by 0.4 °C. The body temperature during sleep is generally stable. At the same time, whether it is awake or sleeping, the fluctuation of body temperature has a certain periodicity. In the awake state, a fluctuation cycle is about 45 min, while in sleep state the cycle is about 90 min. These data help to decide the sleep time and wake-up time, the relationship of falling asleep, the decrease in body temperature and waking up during the rise of body temperature. It can help to reduce the time of falling asleep and the physical discomfort during wake-up.

For most of the people often accompanied by increased metabolic rate and body temperature rises after eating. A healthy body temperature after eating will flatten out and a person with diabetic metabolism system will have abnormal insulin secretion, thus, leading to metabolic imbalances. Long-term monitoring of temperature can help diabetic patients to find a more appropriate diet plan. At the same time, it can be used as reference data for doctors to diagnose. According to the Figure 9a,e, it can be found that body temperature after a 18:30 meal, subjects’ temperatures were steadily rising. After entering the steady state, the body temperature swung around 37 °C, then it began to decline after about 180 min and this is corresponding to the food digestion time. During the digestion temperature, due to the ascension of metabolic rate increased, metabolic decreased after digestion, while the body temperature dropped as well. In Figure 9b,f, in order to explore the relationship between body temperature and eating, the body temperature data from irregular eating in a day were selected for analysis. The lunch included 180 mL milk and 100 g whole wheat bread, then with dinner at 18:30. When the temperature dropped in the evening, the body temperature rose rapidly to a higher level, and kept fluctuating around 37.0 °C. It started to drop after 22:00, which was the digestion time. Accordingly, regular diet is the guarantee of normal body temperature. The abnormal eating will lead to metabolism of body temperature at low level. Consequently, the diet with the effect of reducing weight will be limited, because after the diet, it will reduce the metabolism. Combining with the analysis of the previous section after the motion effect, subjects increasing movement is the first choice of burning fat to reduce weight.

At present, due to the popularity of smart phones, computers and other remote equipment, more and more people formed the habit of staying up late. However, staying up late will cause harm to human body health. The harmful effect appears gradually, such as immune system, liver damage and temperature change. The staying up late trend has not attracted much attention and conditions such as temperature can reflect the body’s metabolism. Figure 9c,g shows the body temperature data of the subjects after staying up late. The visible body temperature is a certain cyclical change, which is in a cycle at first and lowered after waking up. At the same time, when the subject is awake, the temperature is higher than when asleep. The average difference is about 0.4 °C, which is determined by the body’s metabolic rate. The guiding significance of this broken line is that choosing to fall asleep at night when the body temperature starts to drop can help the subject fall asleep faster, corresponding to 3:00 in the figure above, which can improve the quality of sleep. Choosing to wake up in the morning during the rising cycle of body temperature can help reduce the discomfort after waking up, corresponding to 10:00 in the figure above, which can better start the day’s study and work. According to the analysis in Figure 9d,h, after staying up late and getting up from 12:00 to 14:00, the body temperature fluctuates for about 120 min, which is mainly due to going out at noon. After that, the body temperature enters a stable period, which starts at 14:00 and ends at 21:00. The body temperature is constant between 36.5 °C and 37.0 °C. In the stable period, there is also periodic fluctuation, the period is about 45 min and the fluctuation range is between 0.3 °C and 0.4 °C. Body temperature begins to enter the downward range at 21:00 and the metabolic rate of the human body decreases in preparation for sleep. According to all these measured data, if subjects get enough sleep after staying up late, the body clock will shift back overall. In the evening, the body still maintains a high metabolic rate, which explains why it is difficult to adjust the body clock in time after sleeping late and waking up late. After the body adapts to this rhythm, the change usually brings in discomfort. Future studies should look at the relationship between body temperature and movement in-depth, until more data has been obtained through long-time measurement.

## 5. Discussion

In recent years, there have been a number of major sudden infectious disease events at home and abroad, which have triggered increased discussions and thoughts on health management and personal care in the society. After the end of the epidemic, medical health will surely receive more attention. In epidemic prevention and control in the process, some of the temperature measuring methods are still relatively old-fashioned and inefficient. Many of them need close contact of staff and patients to measure body temperature. The process can be improved in many aspects, such as how to improve the efficiency, how to keep a safe distance and check the object and how to reduce cost [37]. With the continuous progress of the society, the discussion about staying up late and eating healthy has gradually attracted attention, and an increased number of people have realized the importance and necessity of real-time monitoring of their physical status. With such a background, this topic focuses on the temperature measurement, temperature measurement with a focus on long-term, non-inductive measurement, day-to-day work measurement, the NFC technology makes the passive wireless patch work for a long time, the thermal conductivity of modified PDMS makes the patch more efficient and more accurately feel body temperature and it can also be used repeatedly after disinfection. To further reduce the cost of temperature measurement, the total price of a single temperature patch can reach less than 2 Yuan (see Table 1). With the application of Android phone, it can realize more functions, such as timing measurement, alarm when the temperature exceeds the set range and temperature data storage. To sum up, the current problems in the field of long-term temperature measurement are summarized as follows:The thermal conductivity of the flexible patch material is low and the conformal contact ability with the skin is poor.Long-term wear comfort is poor and the power supply problem is difficult to solve.The measurement method and data reading means are backward, which is not convenient in the long-term measurement scenario.

**Table 1 sensors-22-09426-t001:** Comparison of various types of thermometers [38].

	Hg/Alcohol Thermometer	Temperature Paste	Smart E- Thermometer	Direct Read E-Thermometer	Bluetooth Temperature Patch	Infrared Thermometer	This Work
Wearing Position	Underarm/ Mouth/Anus	Everywhere	Underarm	Underarm	chest /everywhere	Everywhere	Everywhere
Reading Approach	Direct	Indirect	Indirect (Bluetooth)	Direct /Bluetooth	Bluetooth	Non-touch	Indirect (NFC)
Dimension (mm)	~3 × 3 × 100	~50 × 50 × 2	~100 × 50 × 5	~50 × 50 × 5	~40 × 40 × 10	~300 × 400 × 100	~25 × 40 × 1
Flexibility /Stretchability	N/A	Flexible /not stretchable	Flexible /not stretchable	Flexible /not stretchable	N/A	N/A	Flexible /partially stretchable
Power Supply	Not required	Not required	Li^+^ Battery	Li^+^ Battery	Li^+^ Battery	AA Battery	Not required
Accuracy (~°C)	0.2	1	0.3	0.06	0.2	0.1	0.1
Reusability /Brittleness	Very Good	Disposable	Good (~300 time)	Disposal stick	Good (~500 time)	Very Good	Good (~200 time)
Safety	Potentially Hazard	Very Good	Good	Good	Good	Very Good	Very Good
Durability	N/A	Very Poor	Poor	Poor	Poor	Very Good	Good
Long-time Recording	N/A	N/A	N/A	N/A	N/A	N/A	Yes
Price (RMB/USD)	~10 RMB /1.5 USD	~4 RMB /0.7 USD	~438 RMB /70 USD	~298 RMB /50 USD	~178 RMB /25 USD	~100 RMB /15 USD	~10 RMB /1.5 USD

In order to overcome these disadvantages, the improvement of this topic can be summarized as follows:To overcome the low thermal conductivity of polymer materials by doping method, graphene nanosheets were added into PDMS to form composites and the thermal conductivity was increased by 27.1%.To develop a comfort and convenience of long-term temperature measurement, the patch is designed with hollow-out sandwich structure, abandoning the traditional printed circuit board, using RF field power supply, combined with flexible packaging materials, the patch is flexible, passive wireless, conformal contact with the skin and comfortable for long-term wear.To realize the systematic analyses of the whole-body temperature reading and data collection, which is combined with the specially developed mobile phone application. The temperature recording, alarm over temperature interval and temperature interval command can be written into the patch, making the whole system more advanced and usable.

## 6. Conclusions

With the development of technology, more medical equipment began to the direction of miniaturization, portable, like blood pressure, electrocardiogram (ECG) and so forth. The body temperature as one of the most important indicators of human body health. However, its measuring technique still can be improved in many aspects. Currently, the most challenging thing to realize is its temperature data analysis of the long-term measurement and storage. In the emergence public health events, the body temperature measurement can fluctuate long-term and also be considered an important indicator for daily analysis of their own physical state. The temperature patch designed in this work can be improved in other area, such as in the requirements of multi-parameter measurement. For the part of thermal conductivity improvement, other different types of materials can be further attempted and the mechanism of improving the thermal conductivity of raw materials by mixing carbon nanomaterials. For the patch packaging part, the cycles of uses can be further improved, which can be achieved by improving the welding process and connecting materials. Therefore, the cycles of uses of the patch can reach more than 1000 times. In the data reading and storage part, the interface can be further enhanced in multiple platforms to prepare for more application scenarios. Finally, it could have the possibility of special scenarios in the future. For the perspective of pandemic prevention and control, the temperature patch can enable the combination of the temperature strips and ID card number at the same time. Hence, the patch can reflect temperature situation and everyone’s history to the epidemic prevention, and control points for more data to support [39]. In terms of data reading, the coil can be fabricated large enough to suit in together with the shape of a door. People in and out of the coil only need to pass through the large coil to upload their body temperature data, which is safer for prevention and control workers and subjects. If necessary, with the help of the NFC RF field writing function of the chip, other useful data, such as detection time and detection location, can be written after each test, so as to better prevent and control the epidemic.

In daily life, expansion of the IoT technology on the human body signal collection is attracting more and more attention in scientific research and engineering. Thus, the graphene-enhanced temperature patch can be used in IoT terminal as an important part in the field of health, daily monitoring temperature and to help more people master their natural sleep and body state. It is also a very meaningful thing to help others monitor their health. On the other hand, it will not cause any inconvenience or discomfort in daily life, which supports the hollow, passive wireless, small size and flexible design of the patch.

## Figures and Tables

**Figure 1 sensors-22-09426-f001:**
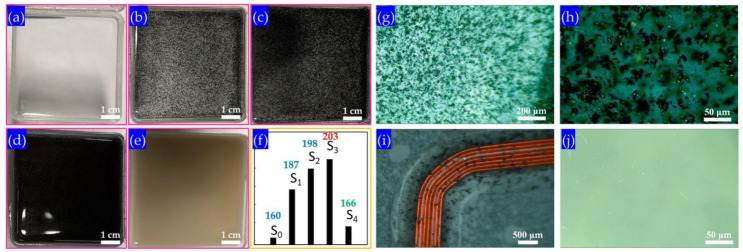
(**a**) S0 sample as-received PDMS membrane after curing; and (**b**) S1 sample with 8.7 wt.% graphene addictive; (**c**) S2 sample with 12 wt.% graphene additives; (**d**) S3 sample with 17 wt.% graphene additives; (**e**) S4 sample with 17 wt.% ND additives; and (**f**) measured thermal conductivity (unit in figure: mW/(m.k)), labeled as S0 to S4, in corresponding from (**a**–**e**), respectively. The detailed optical microscope images were acquired with different magnifications from (**g**–**i**) optical image on 17 wt.% graphene addictive and (**j**) ND addictive PDMS. All the optical images (**g**–**j**) were observed through Nikon LV100POL.

**Figure 2 sensors-22-09426-f002:**
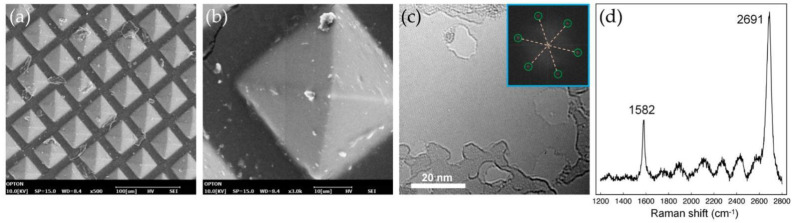
(**a**) Topological SEM image of inverted pyramid structures on graphene-enhanced PDMS; (**b**) Detailed SEM image on inverted pyramid structures with high magnification; (**c**) HRTEM image of the monolayer graphene, inset figure: FFT result of the graphene; and (**d**) Raman results with *G* and 2*D* peaks of the graphene flakes in PDMS.

**Figure 3 sensors-22-09426-f003:**
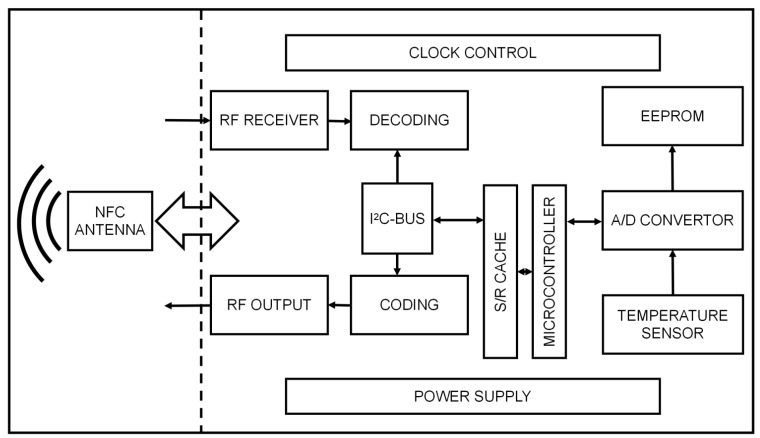
Schematic diagram of electronic configuration on graphene-enhanced PDMS patch for wearable body temperature monitoring. Note that the temperature sensor was integrated with the temperature patch enclosed by PDMS.

**Figure 4 sensors-22-09426-f004:**
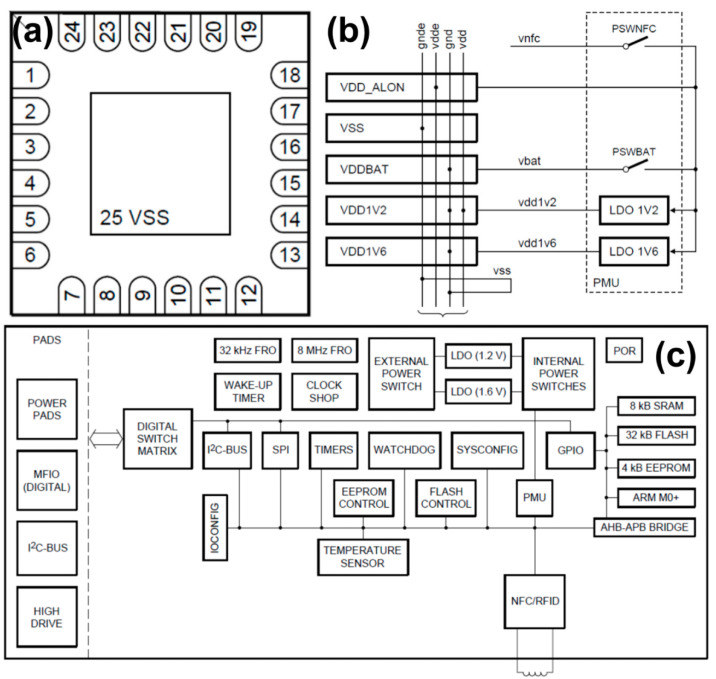
Diagram of NHS3100 with (**a**) chip package with golden relief; (**b**) power supply of the temperature patch connection; (**c**) overall schematic diagram of NHS3100 chip.

**Figure 5 sensors-22-09426-f005:**
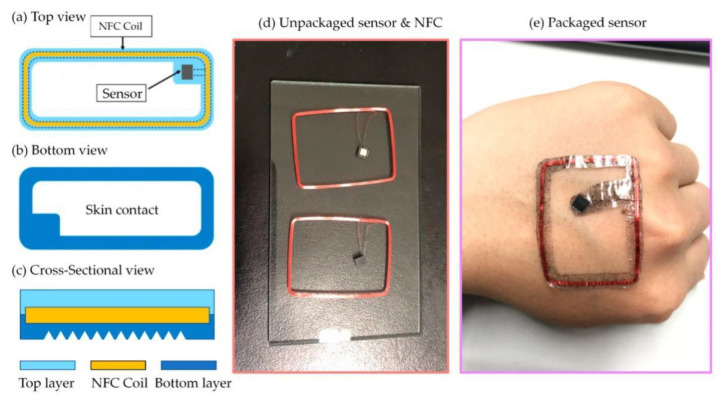
Schematic diagram of the wearable body temperature monitoring device: (**a**) Top view; (**b**) Bottom view; (**c**) Cross-sectional view; (**d**) Unpackaged sensor and NFC copper coil; (**e**) Packaged sensor with graphene-enhanced PDMS layer and its conformal demonstration.

**Figure 6 sensors-22-09426-f006:**
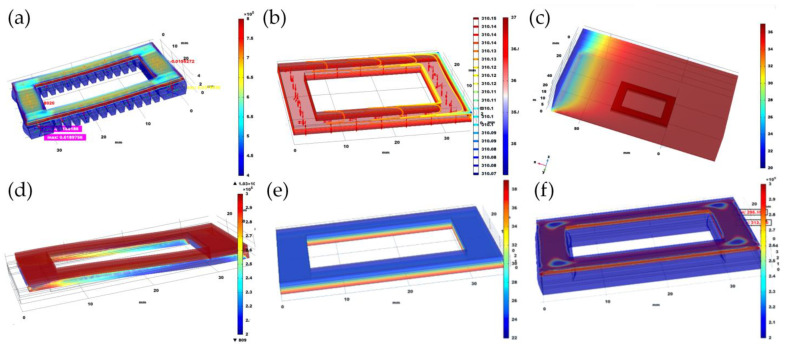
Simulation on COMSOL modelling of the temperature patch: (**a**) basic structure model with deformation of stress distribution contour on the device; (**b**) temperature distribution contour with surface air floats under cover cloth; (**c**) simulated pressure-stress distribution on device on bare skin without cover cloth; (**d**) deformation of patch subjected to 1N force in YZ direction; (**e**) simulation diagram of temperature measurement in bare skin area; (**f**) simulation diagram of stress distribution inside the patch after heating.

**Figure 7 sensors-22-09426-f007:**
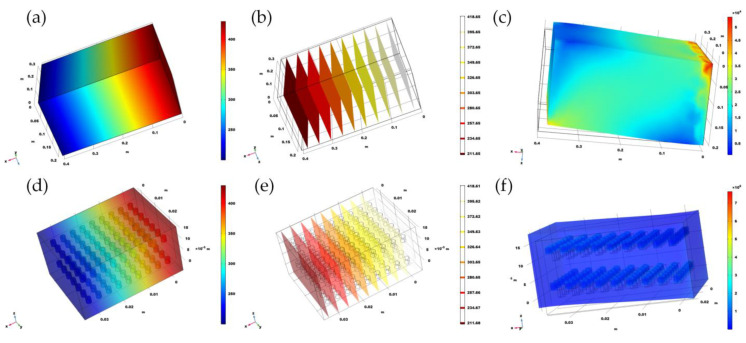
Simulation diagram of thermal isotherms before and after doping: (**a**) Temperature distribution without thermal conductivity particles; (**b**) isotherm distribution without adding thermal conductivity particles; (**c**) stress analysis simulation of PDMS without thermal conductivity particles; (**d**) temperature distribution after doping with thermal conductivity particles; (**e**) isotherm distribution after doping with thermal conductivity particles; (**f**) Stress analysis simulation of PDMS with high thermal conductivity particles.

**Figure 8 sensors-22-09426-f008:**
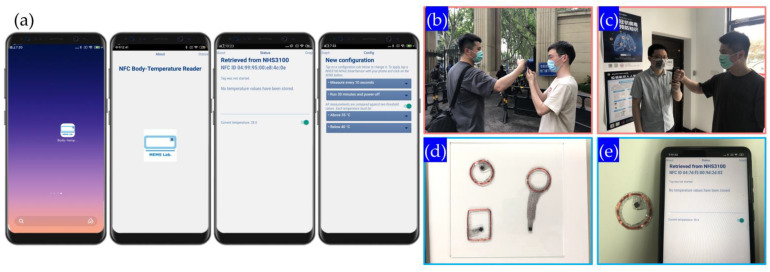
(**a**) Android program for long-term monitoring; application scenarios of (**b**) main entrance of college; (**c**) indoor check-in with real-time/trace-back recording; (**d**) wearable temperature patch enclosed with PDMS; (**e**) wearable temperature patch connecting with Android system in remote mode and its APP interface.

**Figure 9 sensors-22-09426-f009:**
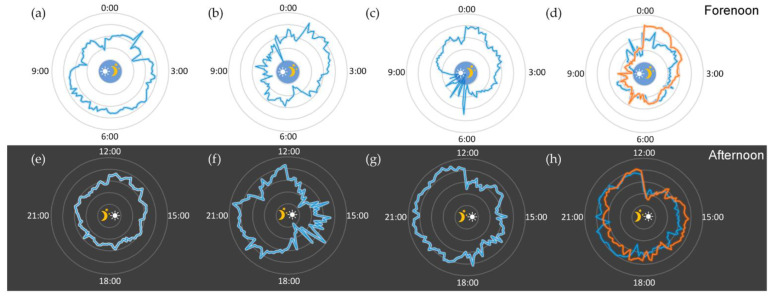
Body temperature recorded by graphene-enhanced PDMS wearable temperature patch: (**a**) forenoon data recorded in regular daily life (1/2 day); (**b**) forenoon data recorded in regular daily life (2/2 day, without diet); (**c**) forenoon data recorded after high-intensity training; (**d**) forenoon data recorded after over-night working (blue line: day 1, orange line: day 2); (**e**) afternoon data recorded in regular daily life (1/2 day); (**f**) afternoon data recorded in regular daily life (2/2 day, without diet); (**g**) afternoon data recorded after high-intensity training; (**h**) afternoon data recorded after over-night working (blue line: day 1, orange line: day 2).

## Data Availability

Not applicable.

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
