# Peer review of "Graphene-Enhanced Polydimethylsiloxane Patch for Wearable Body Temperature Remote Monitoring Application"

_sensors, 2022, doi:10.3390/s22239426_

Round 1

Reviewer 1 Report

This manuscript presents an application of graphene-enhanced PDMS wearable temperature sensor for remote monitoring, which can be utilized in many temperature measurement scenarios under complex public health situations. Comparing with the existing measurement techniques, the remote temperature sensor has many advantages, e.g. wearable, long-time recording, and so forth. It is a topic of interest to the researchers in related area. However, careful revision and minor corrections are required for these specific points:

1. It is noted that your manuscript needs careful editing by someone with expertise in technical English editing paying particular attention to grammar and sentence structure so that the goals and results of the study are clear to the readers. There are a few typos and grammar errors in this manuscript. The English writing still requires improvement.

2. The authors should summarize the main contributions of this manuscript at the end of the first section.

3. The author presented a graphene-enhanced temperature sensor. But the analysis on graphene properties is limited, as well as its contribution to the materials. More analysis and discussion should be added in the manuscript.

4. The figures 1,2,8,9 should be arranged in (a),(b),(c)..., but not for the Capital formats. Details on these figures are required for correction.

5. The testing results (pp.15/19, after Figure 9.) is presented, but its relationship with the healthy is not discussed. More detailed discussions and analysis are required, and its relationship with the body movement can be enhanced.

Author Response

This manuscript presents an application of graphene-enhanced PDMS wearable temperature sensor for remote monitoring, which can be utilized in many temperature measurement scenarios under complex public health situations. Comparing with the existing measurement techniques, the remote temperature sensor has many advantages, e.g. wearable, long-time recording, and so forth. It is a topic of interest to the researchers in related area. However, careful revision and minor corrections are required for these specific points:

  1. It is noted that your manuscript needs careful editing by someone with expertise in technical English editing paying particular attention to grammar and sentence structure so that the goals and results of the study are clear to the readers. There are a few typos and grammar errors in this manuscript. The English writing still requires improvement.

Response: We appreciated the reviewer’s kind suggestion, and we have detailed checked the typos and revised the latest version of manuscript, including the grammar.

  1. The authors should summarize the main contributions of this manuscript at the end of the first section.

Response: We have summarized the main contribution and put it in the introduction part (Line 60 and 77).

  1. The author presented a graphene-enhanced temperature sensor. But the analysis on graphene properties is limited, as well as its contribution to the materials. More analysis and discussion should be added in the manuscript.

Response: Detailed discussion has been added in the paragraph after the characterization of graphene materials. Meanwhile, the contribution from graphene addictive has been explained.(Line 190-205)

  1. The figures 1,2,8,9 should be arranged in (a),(b),(c)..., but not for the Capital formats. Details on these figures are required for correction.

Response: Figures and labels have been rearranged as reviewer's request.

  1. The testing results (pp.15/19, after Figure 9.) is presented, but its relationship with the healthy is not discussed. More detailed discussions and analysis are required, and its relationship with the body movement can be enhanced.

Response: The relationship of healthy, as well as the analysis data has been enhanced in the latest version of manuscript.(Line 633-635)

Reviewer 2 Report

In this paper, the authors designed and demonstrated a temperature sensor using graphene-enhanced PDMS patch for wearable body temperature monitoring with near field communication component. They used graphene nanoplates to improve the thermal conductivity of the patch. There are few questions and comments which I would like the authors address them and make appropriate modifications to the manuscript. I recommend publication of the paper after the authors address the following comments.

1) What is the lateral size, size distribution, and number of layers of the graphene nanoplates? These tend to affect the material property and hence the performance of the sensor.

2) How many sensors did you fabricate? Typically, data from multiple sensors is included to demonstrate the repeatability of their performance.

3) Scale bars are missing in Figure 1.

4) Thermal conductivity values and dimension should be added in Figure 1 (F). How many samples did you make/measure for each wt%?

5) What is the rationale for choosing 8.7 wt%, 12 wt% and 17 wt% graphene?

6) In section 2.2, and at the end of first paragraph “data” has been used twice.

7)  How did you verify the uniform dispersion of graphene nanoplates in PDMS?

Author Response

In this paper, the authors designed and demonstrated a temperature sensor using graphene-enhanced PDMS patch for wearable body temperature monitoring with near field communication component. They used graphene nanoplates to improve the thermal conductivity of the patch. There are few questions and comments which I would like the authors address them and make appropriate modifications to the manuscript. I recommend publication of the paper after the authors address the following comments.

  1. What is the lateral size, size distribution, and number of layers of the graphene nanoplates? These tend to affect the material property and hence the performance of the sensor.

Response: the lateral size of graphene nanoplates varies from 500nm to 300μm, according to our previous investigation. However, the size distribution was difficult to define, for they had different types of aggregations, as well as different layers. The layers of graphene nanoplates can be several to over 10 layers, which could be deduced from the Raman spectra analysis. We have added relevant information and discussion in the paragraph.(Line 200-202)

  1. How many sensors did you fabricate? Typically, data from multiple sensors is included to demonstrate the repeatability of their performance.

Response: Until now, our group have integrated some other sensors in this patch, including ECG, sweat, and others. For the temperature sensing, we have already fabricated more than 10 patches. All of the sensors have shown a very good repeatability during the test. We have added this information in the manuscript. (Line 226-228)

  1. Scale bars are missing in Figure 1.

Response: Figure1 scale bar missing fixed.

  1. Thermal conductivity values and dimension should be added in Figure 1 (F). How many samples did you make/measure for each wt%?

Response: Thermal conductivity values and dimension has been added in Figure 1 (f). We prepared and weighted the graphene flake and sample using electronic balance. Relevant information was provided in the re-submission.

  1. What is the rationale for choosing 8.7 wt%, 12 wt% and 17 wt% graphene?

Response: we chose the different weight percentage for each sample based on our previous research data. To make the graphene flakes distributed in the curing agent solvent homogeneously, the graphene flakes should be mixed in the solution and sent to ultrasonic bath. Thus, the weight percentage was calculated by Weight (graphene flakes)/Weight (graphene flakes + curing agent + PDMS) and this results were optimized based on our previous research articles. We have also attempted higher percentage of graphene in curing agent, but it had significant aggregations in the solvent, which is far from our expectation. We have added the information in the section and also the references. (Line 150-158)

  1. 6) In section 2.2, and at the end of first paragraph “data” has been used twice.

Response: Writing mistakes have been fixed.

  1. How did you verify the uniform dispersion of graphene nanoplates in PDMS?

Response: It is hard to define that the graphene has a uniform dispersion in organic solvent, for it requires quite a long time to observe and test. If the time lasts long enough, the graphene flakes will have aggregation in the organic solvent (curing agents of PDMS) inevitably. So we focused on the final status of graphene distribution in the cured PDMS after annealing and demonstrated the results in Figure 1. We have applied many approaches during the preparation process, such as stirring and ultrasonication, etc. We have added the information in the re-submission manuscript.(Line 150-153)

Reviewer 3 Report

Authors have reported an interesting work on body temperature remote monitoring using Graphene/PDMS patch. Authors are requested to address the following comments

1) Authors can reduce the description of each process. It is always good for the readers if the content is presented precisely instead of dragging it. 

2) Why specific wt.% (8.7, 12, and 17) of graphene additives are considered?

3) How was the optimization obtained at 17 wt.%? what about the thermal conductivities at other wt.% ?

4) Authors are requested to add SEM images of graphene/PDMS nanocomposite and describe its surface morphology.

5) It is required to include the electrical response of each sample (S1 to S4) with respect to temperature variations and find the important parameters like sensitivity and temperature coefficient of resistance (TCR).

Author Response

Authors have reported an interesting work on body temperature remote monitoring using Graphene/PDMS patch. Authors are requested to address the following comments

  1. Authors can reduce the description of each process. It is always good for the readers if the content is presented precisely instead of dragging it.

Response: Thanks for your advice, we will try to reduce some parts of the description in order to demonstrate the whole paragraph precisely.

  1. Why specific wt.% (8.7, 12, and 17) of graphene additives are considered?

Response: we chose the different weight percentage for each sample based on our previous research data. To make the graphene flakes distributed in the curing agent solvent homogeneously, the graphene flakes should be mixed in the solution and sent to ultrasonic bath. Thus, the weight percentage was calculated by Weight (graphene flakes)/Weight (graphene flakes + curing agent + PDMS) and this results were optimized based on our previous research articles. We have also attempted higher percentage of graphene in curing agent, but it had significant aggregations in the solvent, which is far from our expectation. We have added the information in the section and also the references.(Line 137-157)

  1. How was the optimization obtained at 17 wt.%? what about the thermal conductivities at other wt.% ?

Response: As we describe in the last question, it is found that the more graphene flakes added in the patch, the higher thermal conductivities it has. The final thermal conductivities were tested and provided in Figure 1-F. It is found that the sample with 17 wt% GF has the highest thermal conductivity. For higher percentage of graphene flakes, it will significantly aggregated in PDMS and have influence on the final mechanical properties. Thus the maximum amount of graphene addictive was set to 17 wt%.

  1. Authors are requested to add SEM images of graphene/PDMS nanocomposite and describe its surface morphology.

Response: SEM images are shown in Figure 2 with the surface and morphological details. For SEM observation, the graphene in PDMS will not have significant difference in images for it distributes homogeneously inside the PDMS bulk after fully annealing, but not appears on the top surface. Relevant information has been added in the manuscript. (Line 190-200)

  1. It is required to include the electrical response of each sample (S1 to S4) with respect to temperature variations and find the important parameters like sensitivity and temperature coefficient of resistance (TCR).

Response: The information of temperature coefficient of S1 to S4 has been added in the paragraph under Figure 1. However, the electric response was unable to test with respect to the temperature variation. The PDMS patch should be dielectric in order to isolate the sensor within electric signals, as we described in the manuscript.

Round 2

Reviewer 2 Report

Authors addressed my questions and comments. The manuscript is acceptable in the present form.